# *Trichoderma longibrachiatum* Inoculation Improves Drought Resistance and Growth of *Pinus massoniana* Seedlings through Regulating Physiological Responses and Soil Microbial Community

**DOI:** 10.3390/jof9070694

**Published:** 2023-06-21

**Authors:** Cun Yu, Xian Jiang, Hongyun Xu, Guijie Ding

**Affiliations:** 1College of Forestry, Guizhou University, Huaxi District, Guiyang 550025, China; chifengyucun@163.com (C.Y.); gzdxjiangxian@126.com (X.J.); 2College of Eco-Environmental Engineering, Guizhou Minzu University, Guiyang 550025, China; xhyplant@126.com

**Keywords:** drought stress, *Trichoderma longibrachiatum*, *Pinus massoniana*, physiological responses, rhizosphere microbiome

## Abstract

Drought stress poses a serious threat to *Pinus massoniana* seedling growth in southern China. *Trichoderma* species, as beneficial microorganisms, have been widely used in agriculture to enhance plant growth and drought tolerance, but the interaction mechanisms remain unclear. To investigate the effect of drought-resistant *Trichoderma longibrachiatum* inoculation on *P. massoniana* growth under drought stress, the plant physiological indicators and rhizosphere microbiome diversity were measured to identify *Trichoderma*-activated mechanisms. *Trichoderma longibrachiatum* inoculation significantly promoted *P. massoniana* growth under drought treatment, and enhanced nitrogen, phosphorus, and potassium absorption compared with those of non-inoculated seedlings. *Trichoderma longibrachiatum* treatment alleviated the damage to cell membranes and needle tissue structure, and significantly increased antioxidant enzyme activities, osmotic substance contents, and photosynthesis in *P. massoniana* in response to drought stress. Soil nutrient contents, activities of sucrase, phosphatase, and urease as well as the relative abundances of the dominant genera *Burkholderia*, *Rhodanobacter*, and *Trichoderma* were elevated in the rhizosphere soil of *P. massoniana* inoculated with *T. longibrachiatum* under drought stress. A network analysis showed that certain crucial dominant taxa driven by *T. longibrachiatum* inoculation, including *Penicillium*, *Trichoderma*, *Simplicillium*, *Saitozyma*, *Burkholderia*, *Bradyrhizobium*, *Sinomonas*, and *Mycobacterium*, had more correlations with other microorganisms in the soil. *Trichoderma longibrachiatum* enhanced *P. massoniana* seedling growth under drought stress by regulating physiological responses and soil microbial community.

## 1. Introduction

Masson pine (*Pinus massoniana* Lamb.) is an important pioneer tree species used for timber and afforestation in China owing to its rapid growth, high yield, and oleoresin production [1]. It is one of the most important dominant forest species in Guizhou Province [2]. However, within the distribution range of *P. massoniana* in southern China, the rainfall is uneven and accompanied by frequent seasonal droughts, which severely restrict the development of *P. massoniana* timber forests [3]. Especially in the karst area of Guizhou Province, because of the combined effects of the typical monsoon climate, specialized landform, and hydrogeological structure, the soils are prone to drought, which strongly impacts the growth of *P. massoniana* [4]. Currently, the predominant means of improving the drought tolerance of forest seedlings include genetic engineering, breeding for drought resistance, and chemical fertilizer application [5,6,7]. Such approaches may be slow acting or cause environmental pollution. An alternative strategy is to identify active beneficial microorganisms, which are natural, environmentally safe, and more rapidly effective, to enhance the stress resistance of plants.

Recent studies have shown that ectomycorrhizal fungi, such as *Suillus placidus*, *Suillus bovinus*, and *Sclerderma citrinum*, can alleviate the effects of drought stress on *P. massoniana* by establishing a symbiotic relationship, expanding the root absorption area, and improving water use efficiency and nutrient metabolism [8,9]. However, the application of ectomycorrhizal fungi is susceptible to environmental factors and the fungal species grow slowly, usually requiring incubation for 20–30 days before inoculation [9]. In addition, the colonization period is protracted. *Trichoderma* spp. are the most promising beneficial microbial species because they grow rapidly and are strongly tolerant of diverse stresses. *Trichoderma* spp. strongly interact with *Pinus* spp. and can survive in the rhizosphere soil or colonize the roots, and promote the growth of *Pinus* spp. as well as their tolerance to various environmental stresses, including drought, salt, heavy metals, and diseases [10,11,12,13]. However, whether *Trichoderma* spp. are capable of improving the growth and tolerance to drought stress of *P. massoniana* requires further study.

Drought stress leads to the accumulation of reactive oxygen species (ROS), causing oxidative damage to the cell structure, and physiological and metabolic disruption [14]. *Trichoderma* spp. can alleviate the damage of drought stress to plants by a variety of mechanisms, including improving nutrient uptake and photosynthesis and altering antioxidant and proline metabolism [15,16]. For instance, to improve the drought tolerance of *Astragalus mongholicus*, *T. afroharzianum* and *T. longibrachiatum* inoculation enhance root biomass and length by regulating antioxidant enzyme activities, growth hormones, and the ascorbic acid–glutathione redox system [17]. Dual inoculation with *T. viride* and *Paraboeremia putaminume* strongly affects the rhizosphere microbiome [18]. In our previous research, four endophytic *Trichoderma* strains were isolated from *P. massoniana*, and their drought tolerance was evaluated [19]. The *T. longibrachiatum* strain showed the strongest drought resistance, but the mechanism by which it influences *P. massoniana* growth in response to drought stress is unclear.

Based on the aforementioned findings, it was hypothesized that the isolated *T. longibrachiatum* strain would enhance the drought tolerance and growth of *P. massoniana* seedlings. To verify this hypothesis and clarify the mechanism involved, the aims of the present study are (1) to evaluate the effects of *T. longibrachiatum* inoculation on *P. massoniana* seedling growth and physiological indicators, including nutrient uptake, photosynthesis, antioxidant metabolism, and needle tissue structure, under drought stress, and (2) to determine the changes in soil physicochemical properties and rhizosphere microbiome composition. The results will provide insight into the potential of *T. longibrachiatum* application for improving *P. massoniana* seedling growth under drought stress.

## 2. Materials and Methods

### 2.1. Plant Materials, Inoculation Assay, and Drought Treatment

*Pinus massoniana* seeds were collected from the Ma’anshan National Forest Variety Base located in Duyun City, Guizhou Province, China (26°15′ N, 107°30′ E). After soaking in water at 30 °C overnight, the seeds were sterilized with 0.4% KMnO_4_ and 75% ethanol for 30 min and 1 min, respectively. The seeds were then sown in sterilized vermiculite and incubated at 28 °C for 15 days. Subsequently, the seedlings were transferred to plastic trays filled with a mixed medium (Masson pine forest soil:pearlite:vermiculite, 6:3:1 [*v*/*v*/*v*]). Masson pine forest soil was taken from the *P. massoniana* plantation at Guizhou University (26°26′58.47″ N, 106°39′9.01″ E). In order to rule out the possible interference by original microorganisms in the rhizosphere soil of *P. massoniana* seedlings, the mixed medium for plant growth, including soil, pearlite, and vermiculite, was sterilized at 121 °C for 30 min as described by Halifu et al. [13] and Li et al. [17]. The physicochemical properties of the mixed medium were pH 5.92 ± 0.04, total nitrogen 0.59 ± 0.04 g·kg^−1^, total phosphorus 0.95 ± 0.03 g·kg^−1^, total potassium 18.26 ± 0.21 g·kg^−1^, available nitrogen 73.26 ± 1.74 mg·kg^−1^, available phosphorus 17.39 ± 0.63 mg·kg^−1^, and available potassium 326 ± 5.27 mg·kg^−1^. Each tray contained 32 holes (hole depth 110 mm × upper diameter 58 mm × lower diameter 20 mm), and each hole contained three or four seedlings. The seedlings were kept under day/night room temperatures of 23/18 ± 2 °C, a relative humidity of 70 %, an irradiance of 400 μmol m^−2^s^−1^, and a 14 h light/10 h dark photoperiod.

The *T. longibrachiatum* strain (GenBank accession no. MT131278.1) was isolated from the roots of healthy *P. massoniana* trees and stored in the Forest Protection Laboratory, Guizhou University, China. Three pieces of activated *T. longibrachiatum* plugs (diameter 6 mm) were inoculated into a 250 mL triangular flask containing 150 mL potato dextrose broth (PDB) culture medium, and shaken at 120 rpm at 28 °C for 4 days. Then the number of spores was measured by Hemocytometer (XB-K-25, Neubauer, China), eventually adjusting to a spore suspension of 1 × 10^6^ CFU/mL [20]. In a pot experiment, 3-month-old *P. massoniana* seedlings were inoculated with 30 mL *T. longibrachiatum* spore suspension using the root irrigation method [21]. After 30 days, inoculation of the seedlings with *T. longibrachiatum* was repeated. An equal volume of sterilized PDB medium lacking the fungus was applied to non-inoculated seedlings as a control. After inoculation, the growth conditions of the seedlings were the same as before. 

At 60 days after inoculation, the seedlings inoculated with *T. longibrachiatum* (+T) and non-inoculated seedlings (−T) were used for drought-stress treatment. Three watering treatments were applied: well-watered (CK; RS = 75–80%), light drought (LD; RS = 50–45%), and severe drought (SD; RS = 20–15%), determined as RS % = SW/MF × 100, where RS is the relative humidity of the soil, SW is the soil water content, and MF is the maximum field moisture capacity. Each treatment was established with three replicates, and each replicate comprised 60 seedlings. At 15 days after treatment, the seedlings were harvested for observation of root colonization and determination of physiological indicators, and the rhizosphere soil was collected for measurement of soil physicochemical properties, soil enzyme activities, and microbial community analysis. 

### 2.2. Microscopic Observation of Root Colonization 

*Trichoderma* colonization of the roots of *P. massoniana* seedlings was determined using the method described by He et al. [22]. Roots of 30 seedlings inoculated with *T. longibrachiatum* and grown under the three drought treatments were randomly selected. After washing with distilled water five times, the roots were cut into segments of approximately 1 cm in length, soaked in a 10% KOH solution, and heated at 90 °C for 60 min. The samples were washed with distilled water three times, then softened in a hydrogen peroxide solution for 10 min. After softening, the roots were soaked with 2% HCl for 30 min. Finally, the root samples were placed in 0.05% trypan blue solution and heated at 90 °C for 20 min. A total of 100 root samples were randomly selected for observation with a light microscope (Olympus CX21, Tokyo, Japan). The percentage root colonization (PRC) was calculated with the formula PRC (%) = (number of infected root segments/total number of microscopically examined root segments) × 100. 

### 2.3. Measurement of Seedling Relative Water Content and Growth Variables 

The seedlings were harvested and weighed immediately to determine the fresh weight (Wf). The seedlings then were soaked in distilled water until attaining the saturated fresh weight (Ws). Subsequently, the seedlings were dried at 80 °C to a constant weight to record the dry weight (Wd). The seedling relative water content (RWC) was calculated with the formula RWC (%) = [(Wf − Wd)/(Ws − Wd)] × 100 [15]. Individual needles were weighed immediately to determine the initial fresh weight (FW). The needles were then spread on clean filter paper for natural evaporation and weighed at 30-min intervals to determine the desiccated weight at different time points (FWt). Finally, the needles were dried at 80 °C to obtain the dry weight (DW). The water loss rate (WLR) was calculated as WLR (%) = [(FW − FWt)/(FW − DW)] × 100 [23].

Plant shoots and roots were separately harvested and weighed. The root morphology was scanned with a desktop scanner (EPSON Perfection V800 Photo; Epson, Nagano, Japan). The WinRHIZO software (version 2009a, Regent Ltd., Quebec, QC, Canada) was used to measure the total root length, root surface area, root volume, and ground diameter of the root system.

### 2.4. Tissue Structure Observation

Needles were cut into segments of 1–2 mm in length, which were embedded in paraffin and sectioned (approximately 8 μm thickness) with a microtome (RM2016, Shanghai Leica Instruments Co, Ltd., Shanghai, China). The sections were rehydrated in BioDewax and Clear Solution I, then BioDewax and Clear Solution II, for 20 min each, then sequentially soaked in 100% ethanol and 75% ethanol for 5 min each, and then rinsed with tap water. Subsequently, the sections were stained with safranin solution for 2 h, rinsed with water, then decolorized with 50%, 70%, and 80% ethanol for 3–8 s. The sections were placed in plant solid green staining solution for 6–20 s, then xylene for 5 min, and were observed with a microscope imaging system (Nikon DS-U3, Nikon, Tokyo, Japan).

### 2.5. Measurement of Osmolytes and Antioxidant Enzyme Activities

Fresh seedlings were used for the measurement of osmolyte contents and antioxidant enzyme activities. Soluble sugar and soluble protein contents were determined using the anthrone colorimetric and Bradford methods [24]. The fresh samples were extracted with 3% aqueous sulfosalicylic acid and reacted with ninhydrin to measure the proline content [25]. The thiobarbituric acid method was used to assay the malondialdehyde (MDA) content [26]. Relative electrical conductivity was measured using the method described by Fan et al. [27]. Chlorophyll was extracted with 80% (*v*/*v*) acetone and the absorbance was measured at 663 and 645 nm [28]. The activities of superoxide dismutase (SOD), peroxidase (POD), catalase (CAT), ascorbate peroxidase (APX), glutamine synthetase (GS), and glutamate synthetase (GOGAT) were determined using assay kits (Grace Biotechnology Co., Ltd., Suzhou, China). 

### 2.6. Photosynthesis Analysis

Nine seedlings from each treatment were randomly chosen to measure chlorophyll fluorescence and photosynthetic indicators. The seedlings were dark-adapted for 30 min, then a portable chlorophyll fluorometer (Cl-340, WALZ, Wurzburg, Germany) was used to assay the initial fluorescence (*F*_0_) and the maximum fluorescence (*F*_m_) as described by Li et al. [9]. The chlorophyll fluorescence (*F*_v_/*F*_m_) value was calculated with the formula *F*_v_/*F*_m_ = (*F*_m_ − *F*_0_)/*F*_m_ [29]. Intercellular CO_2_ concentration (*C*_i_), transpiration rate (*T*_r_), net photosynthetic rate (*P*_n_), and stomatal conductance (*G*_s_) were measured on a sunny morning using a Li-6800xt (LI-COR, Lincoln, NE, USA). The gas exchange parameters were set as described previously [11]. Water-use efficiency (WUE) was calculated with the formula WUE = *P*_n_/*T*_r_.

### 2.7. Determination of Nutrient Elements in Seedlings and Rhizosphere Soil

The dried seedlings and rhizosphere soil samples were digested with H_2_SO_4_ and HClO_4_. The semi-micro Kelvin, molybdenum antimony colorimetric, and flame photometric methods were applied to determine the total nitrogen (TN), total phosphorus (TP), and total potassium (TK) contents, respectively. Available nitrogen (AN) content in the soil was determined with the alkaline hydrolysis diffusion method. Sodium bicarbonate was used to extract available phosphorus (AP) in the soil, then the molybdenum blue method was applied to measure the AP content. Available potassium (AK) in the soil was extracted with ammonium acetate and quantified by flame photometry. Soil urease, phosphatase, and sucrase activities were measured using the methods described by Akhtar et al. [30]. Air-dried soil samples (0.5 g) were passed through a 0.25 mm sieve and digested with 0.4 mol/L K_2_Cr_2_O_7_–H_2_SO_4_ solution, then titrated with FeSO_4_ to measure the organic matter content in the soil. 

### 2.8. DNA Extraction, High-Throughput Sequencing, and Bioinformatic Analysis

After drought treatment for 15 days, 18 rhizosphere soil samples (from the 6 treatments, with 3 replicates per treatment) were harvested for DNA extraction using the FastDNA^®^ Spin Kit for Soil (MP Biomedicals, Irvine, CA, USA). Samples were determined for DNA purity and concentration using 1% agarose gel electrophoresis and QuantiFluor™ -ST (Promega, Madison, WI, USA). Quantified DNA samples were used to identify the microbial community by amplifying the internal transcribed spacer (ITS) region in fungi and the 16S rRNA gene in bacteria using the primer pairs ITS1/ITS2 and 338F/806R, respectively [31]. The purified PCR amplicons were sequenced using an Illumina MiSeq PE300 platform (Illumina Inc., San Diego, CA, USA). Paired-end reads were quality-filtered and merged as raw tags using FLASH (version 1.2.7, Magoc, T. USA). The 16S rRNA and ITS sequences were clustered into operational taxonomic units (OTUs) with a 97% similarity cutoff using UPARSE software (version 7.1, California, USA) [32]. A representative sequence with the highest abundance was selected for each OTU. The sequences of fungi and bacteria were taxonomically assigned based on the UNITE and SILVA databases, respectively [33]. The relative abundance of each phylum or genus was calculated using the following formula: relative abundance (%) = *n_i_*/*N*, where *n_i_* is the number of sequences for each OTU, *i* represents an individual OTU, and *N* is the total number of sequences for all OTUs in the sample. The richness and diversity of microbial composition were determined by calculating α-diversity indices, comprising the Chao, abundance-based coverage estimator (ACE), Shannon, and Simpson indices, using the vegan package for R software (version 2.5.4, Boston, MA, USA). Principal component analysis (PCA) was conducted to visualize the compositional differences in the bacterial or fungal community structure under the different treatments using the Majorbio Cloud Platform. 

### 2.9. Statistical Analysis

Statistical analyses were performed with IBM SPSS Statistics 21.0 software (IBM, Armonk, NY, USA). A one-way analysis of variance (ANOVA) was used to analyze the significance of differences in the various physiological and soil physicochemical variables with Duncan’s multiple range test (*p* ≤ 0.05). A two-way ANOVA was applied to analyze the statistical significance (α = 0.05) of *Trichoderma* inoculation and the drought treatments on the physiological and growth indicators. The OTUs with a relative abundance of more than 0.01% in each sample were chosen to construct a co-occurrence network, and dominant taxa were screened based on a relative abundance greater than 0.1% in each sample. Based on Spearman’s correlation coefficients (*R* > 0.6 or *R* < −0.6; *p* < 0.05), the bacterial and fungal co-occurrence network was visualized with Gephi software (version 0.9.3, Paris, France). Spearman correlation coefficients were calculated to assess the pairwise relationships between soil physicochemical properties and microbial community composition.

## 3. Results

### 3.1. Trichoderma longibrachiatum Colonization of P. massoniana Roots Reduced Seedling Water Loss under Drought Stress

To observe the colonization of *T. longibrachiatum* after inoculation, spores and hyphae structures were observed in all stained root segments. As shown in Figure 1A, there were no germinated spores and hyphae observed in the roots of seedlings under CK/−T treatment. However, they did emerge under CK/+T, LD/+T, and SD/+T treatments, indicating *T. longibrachiatum* could successfully colonize the roots of *P. massoniana* seedlings after inoculation for 60 d. With an increase in the severity of drought stress, the root colonization rate progressively declined and attained 53.52% under SD/+T treatment, which represented a reduction of 25.32% compared with that of CK/+T treatment (Figure 1B).

The water loss of seedlings was quantified under different drought stresses. No significant difference in relative water content (RWC) of the needles was observed between the *Trichoderma* inoculation (+T) and *Trichoderma* non-inoculation (−T) treatments under the well-watered treatment (CK), whereas *Trichoderma* inoculation significantly increased the RWC under the light drought (LD) and severe drought (SD) treatments compared with that of non-inoculated seedlings (Figure 1C). In addition, *Trichoderma* inoculation decreased the water loss from needles; in particular, the water loss rate in the SD/+T treatment was significantly lower than that in the SD/−T treatment (Figure 1D). 

### 3.2. Trichoderma longibrachiatum Promoted the Growth and Nutrient Absorption of P. massoniana Seedlings in Response to Drought Stress 

To verify the effect of *T. longibrachiatum* inoculation on seedling growth under drought stress, the growth and nutritional indexes of seedlings were measured. The seedlings gradually wilted under drought stress, and wilting was more severe under the SD/−T treatment than the SD/+T treatment (Figure 2A). A two-way ANOVA revealed that the *Trichoderma* and drought treatments significantly affected the growth and nutrient uptake by the seedlings (*p* < 0.05). Most growth indicators and nutrient contents measured decreased under the LD and SD treatments; however, *Trichoderma* inoculation (+T) significantly improved seedling height, root volume, shoot fresh weight, and phosphorus, potassium, and nitrogen contents under drought stress compared with those of the −T treatment (Figure 2B–M).

### 3.3. Drought-Induced Damage to the Needle and Root Tissues Were Alleviated by T. longibrachiatum Inoculation 

To evaluate the damage to cell membranes and tissue structure caused by drought stress, the needle MDA content and relative electrical conductivity (REC) were measured and the needle anatomy was observed (Figure 3). Compared with the CK, the LD and SD treatments increased the MDA content and REC (Figure 3A,B), implying that membranes suffered increased damage with increased drought severity. However, seedlings inoculated with *T. longibrachiatum* (+T) had lower MDA contents and REC compared with those of non-inoculated seedlings (−T) under the CK, LD, and SD treatments (Figure 3A,B).

Under the CK/−T and CK/+T treatments, the overall tissue structure of the needle was intact and the cell size was uniform; mesophyll cells were abundant and evenly distributed between the epidermis and endodermis, and the endodermal cells were evenly, tightly, and orderly arranged (Figure 3C). Under the LD/−T and LD/+T treatments, the relative abundance of mesophyll cells was reduced, and the endodermis was slightly shrunken and deformed compared with the CK treatment (Figure 3C). The needle structure differed notably between the +T and −T treatments under severe drought stress. The needle structure, including the mesophyll cells, endodermis, and vascular tissues, was severely deformed and shrunken, the mesophyll cell membranes were damaged and dispersed, and the cells surrounding the resin canals are almost invisible in the SD/−T treatment, whereas the cell structure was relatively intact with only minor defects observed under the SD/+T treatment (Figure 3C).

### 3.4. Trichoderma longibrachiatum Treatment Changed Osmolytes and Antioxidant Metabolism in P. massoniana under Drought Stress

To further investigate the effect of *T. longibrachiatum* inoculation and drought stress on the seedling physiology, antioxidant and osmotic adjustment abilities were evaluated. A two-way ANOVA showed that osmolyte accumulation and antioxidant metabolism were markedly affected by the *T. longibrachiatum* and drought treatments (*p* < 0.05; Figure 4). Drought stress caused an increase in the contents of osmotic substances (including soluble sugar, soluble protein, and proline contents) and antioxidant enzyme activities (including SOD, POD, CAT, and APX) compared with those of the CK treatment; in addition, values for these indicators were higher in the +T treatment than in the −T treatment under the same watering treatment (Figure 4A–G). Glutamine and GOGAT activities were significantly reduced with an increase in drought-stress severity, whereas the activities were significantly higher in the +T treatment than in the −T treatment under all watering treatments (Figure 4H,I). 

### 3.5. Drought-Induced Photosynthetic Inhibition Was Mitigated by T. longibrachiatum Inoculation

The drought and *T. longibrachiatum* treatments significantly affected the photosynthesis of *P. massoniana* seedlings (*p* < 0.05; Figure 5). Under the LD and SD treatments, various photosynthetic indicators (including *F*_v_/*F*_m_, *T*_r_, *P*_n_, *C*_i_, and *G*_s_) and the total chlorophyll content declined (Figure 5A–E and Appendix A). However, the values of these photosynthetic indicators were significantly higher in the +T treatment than in the −T treatment under the CK, LD, and SD treatments. The WUE was elevated under the LD and SD treatments, but the difference was not significant between the −T and +T treatments (Figure 5F).

### 3.6. Rhizosphere Soil Nutrient Contents and Enzyme Activities Were Improved by T. longibrachiatum Inoculation

To verify how *T. longibrachiatum* affects rhizosphere soil fertility, thereby regulating seedling drought resistance, the soil nutrient contents and enzyme activities of *P. massoniana* seedlings were measured. Except for the AN and OM contents, the soil nutrient indicators were deceased under the LD and SD treatments (Appendix A). In contrast, the activities of sucrase, phosphatase, and urease were enhanced under drought stress (Appendix A). While, *T. longibrachiatum* treatment markedly improved soil nutrient contents and enzyme activities compared with those of the −T treatment under the CK, LD, and SD treatments (Appendix A). A correlation analysis showed that significant positive correlations existed between most nutritional indices; available K and P had significant positive correlations with OM, TN, TK, and TP (Appendix A). Sucrase and phosphatase had a significant positive correlation with AN; while significant negative correlations were observed between soil enzyme activities and most soil nutrient indicators (Appendix A).

### 3.7. Composition of the Rhizosphere Soil Microbial Community 

The rhizosphere soil microbial communities induced by *T. longibrachiatum* under different drought conditions were measured. The composition of the soil bacterial and fungal communities differed among the drought and *Trichoderma* treatments as indicated by PCA (Appendix A). Among the α-diversity indices for fungi, the Shannon, ACE, and Chao index values in the SD/+T treatment were significantly improved compared with those of the SD/−T treatment, whereas the values of these indices for bacteria were decreased, indicating that *T. longibrachiatum* treatment improved the richness and diversity of the fungal community under severe drought stress, but those of the bacterial community were reduced (Appendix A). Proteobacteria, Actinobacteriota, Acidobateriota, and Bacteroidota were the predominant bacterial phyla detected in all soil samples. The relative abundances of Proteobacteria were greater in the +T treatment than in the −T treatment under the CK, LD, and SD treatments (Figure 6A). Ascomycota and Mortierellomycota were the predominant fungal phyla in the soil samples. The relative abundance of Ascomycota was reduced in the LD/+T and SD/+T treatments compared with that of the other treatments, whereas the abundance of Mortierellomycota was significantly increased (Figure 6B). At the genus level for bacteria, *Burkholderia* and *Rhodanobacter* were the predominant genera detected, the abundances of which were enhanced under the +T treatment (Figure 6C). At the genus level for fungi, the relative abundance of *Trichoderma* was elevated in the +T treatment, whereas that of *Penicillium* and *Talaromyces* was reduced (Figure 6D).

In addition, the relationships between soil physicochemical properties and the 30 most abundant genera were analyzed (Appendix A). The TK, TP, AK, TN, and AP contents were mainly positively correlated with norank_f_Caulobacteraceae. *Burkholderia* and *Rhodanobacter* had a significant positive correlation with AN content. Most fungal genera were significantly positively correlated with Ph, Ur, Su, OM, and AN content. *Trichoderma* had a significant positive correlation with OM, AN, and Ur. 

### 3.8. Co-Occurrence Network Analysis 

To further understand the interaction between microorganisms, co-occurrence networks were constructed with the nodes represented by OTUs (Figure 7). The number of nodes, edges, and dominant taxa among bacteria were significantly higher than those for fungi, indicating that the network structure for bacteria was more complex than that for fungi. Under the −T and +T treatments, 11 and 14 dominant fungal taxa were identified, respectively. The dominant fungal taxa under the −T treatment, comprising OTU153, OTU156, OTU38, and OTU514, which belonged to the genera *Penicillium*, *Talaromyces*, and *Trichoderma*, were closely associated with other microorganisms (Figure 7A). The dominant fungal taxa under the +T treatment, comprising OTU488, OTU153, OTU180, OTU312, OTU342, and OTU97, which belonged to the genera *Penicillium*, *Trichoderma*, *Simplicillium*, and *Saitozyma*, showed close interactions with other microorganisms (Figure 7B). With regard to bacteria, the average number of dominant taxa in the +T treatment was 26, compared with 21 in the −T treatment, indicating that *T. longibrachiatum* treatment enhanced the interaction between the dominant bacterial taxa and other microorganisms. In the −T and +T treatments, 60 and 56 dominant bacterial taxa were identified, respectively. Regarding the −T treatment, certain important OTUs (e.g., OTU155, OTU136, OTU1415, and OTU1443), which had a relatively large degree, mainly belonged to the genera norank_f__Isosphaeraceae, norank_f__Micropepsaceae, *Frateuria*, and *Leifsonia* (Figure 7C). With respect to the +T treatment, certain crucial OTUs (OTU105, OTU501, OTU335, OTU1155, OTU418, OTU578, and OTU1029), which had a relatively large degree, mainly belonged to *Burkholderia*, unclassified_f__Comamonadaceae, *Bradyrhizobium*, *Sinomonas*, *Mycobacterium*, and norank_f__Micropepsaceae (Figure 7D).

## 4. Discussion

The application of beneficial microorganisms can not only enhance soil fertility and promote plant growth but also improve the tolerance of plants in response to various biotic and abiotic stresses [34]. *Trichoderma*, a common and widely distributed fungal genus, is frequently used in agriculture but less in tree seedling cultivation. Previous studies have shown that *T. longibrachiatum* can improve plant salinity tolerance [35,36], but research into its effects on plant drought resistance is limited. In the present study, *T. longibrachiatum* successfully colonized the roots of *P. massoniana* under drought stress and reduced the water loss rate of the seedlings. Similar phenomena have been observed for other *Trichoderma* spp.; for instance, treatment with *T. asperellum* strain T34 increases the leaf RWC of maize under drought stress [15]. In addition, *T. longibrachiatum* increased the seedling height, root volume, and root fresh and dry weights as well as the seedling nitrogen, phosphorus, and potassium contents under drought stress compared with those of non-inoculated seedlings. *Trichoderma* inoculation strongly affects root system development, which is associated with an increase in plant yield; for instance, *T. virens* produces auxin-related compounds that activate auxin-regulated gene expression in *Arabidopsis* and promote lateral root growth [37]. Some *Trichoderma* species have a strong capability for rhizosphere colonization and the mycelia can grow in association with the plant root system; the mycelia effectively utilize complex carbohydrates in the soil as carbon sources and increase the nutrient availability for plant growth to enhance stress tolerance [13,38]. *Trichoderma* treatment stimulates the carbon and nitrogen metabolism of *Codonopsis pilosula* [39]. The nitrogen and sulfur contents are increased in drought-stressed sugarcane treated with *T. asperellum* [16]. The present results are consistent with these findings, and thus it was speculated that *T. longibrachiatum* might effectively enhance drought tolerance by regulating root development and stimulating nutrient metabolism in *P. massoniana*. 

Drought stress can disrupt crucial physiological and biochemical activities in plants, causing oxidative stress, damaging tissue structure, and disrupting membranes and macromolecules [40]. The REC and MDA contents are indicators of the degree of membrane permeability and lipid peroxidation under abiotic stress [41]. In the present study, although drought stress led to increases in REC and MDA contents in the −T and +T treatments, the increase was significantly lower in the +T treatment. These results suggested that *Trichoderma* treatment reduced drought-induced damage to cell membranes, which was consistent with the findings of Shukla et al. [42]. In addition, *T. longibrachiatum* reduced the severity of damage to the needle mesophyll cells, endodermis, and vascular tissues under drought stress. Plant–microbe interactions have a critical influence on plant metabolism, including altering osmolyte accumulation, enzyme activities, and phytohormone contents, thereby alleviating plant cell damage in response to stress exposure [43]. The accumulation of osmolytes assists with osmotic adjustment to maintain cell hydration and membrane structural integrity in response to a water deficit [44]. Proline accumulation in the cytoplasm is strongly associated with the ionic balance in the vacuoles [45]. An increase in osmolyte content has been reported in various crops in response to treatment with beneficial microbes [46], consistent with the present results. *Trichoderma longibrachiatum* treatment enhanced the accumulation of proline, soluble sugar, and soluble protein under drought stress, which may play important roles in regulating the osmotic balance of *P. massoniana* to reduce water loss and maintain anatomical stability. Antioxidant enzymes eliminate ROS, which otherwise readily trigger oxidative damage in plant cells [47]. Inoculation of certain beneficial microbes, such as *Arthrobacter* spp., *Bacillus* spp., and *Pseudomonas* spp., promotes antioxidant enzyme activities in plants under drought stress [48]. The present results showed that *T. longibrachiatum* inoculation induced higher activities of SOD, POD, CAT, APX, GS, and GOGAT compared with those of non-inoculated seedlings under drought stress, indicating that *T. longibrachiatum* treatment alleviated drought-induced injury in *P. massoniana* by enhancing the ROS-scavenging capacity.

Under drought stress, the accumulation of ROS causes damage to the photosystems, leading to photosynthetic inhibition [49]. Excessive ROS can impair the chloroplasts, decrease the photochemical reactions, and finally suppress the photosynthesis and yield of the crop [50]. Plants have developed several enzymatic antioxidant protection mechanisms to counteract the damaging effects of ROS under environmental stresses. Photosystems also could evolve a highly efficient antioxidant defense system, including enzymatic scavengers (SOD and CAT), reducing ROS accumulation [51]. In addition, drought stress reduces chlorophyll content, which also impacts photosynthesis [52]. Drought stress could disrupt tissue structure, leading to cell deformation, rupture, and even death. After cell death, chlorophyll would be released from the chloroplast, and free chlorophyll was unstable and easily degraded [53]. Recent studies indicate that *Trichoderma* spp. can reduce ROS accumulation and degradation of photosynthetic pigments, and thus enhance photosynthetic activity under drought stress [16]. The present results are consistent with these findings, and thus *T. longibrachiatum* enhanced tolerance to drought by enhancing the photosynthetic capacity of *P. massoniana* seedlings. 

Under exposure to drought stress, the interaction between *T. longibrachiatum* and *P. massoniana* not only induced a plant physiological response but also altered the rhizosphere environment. Drought stress reduces the soil moisture content and plant transpiration rate, which rapidly lead to a reduction in soil nutrient availability and nutrient uptake by plants, and restricts plant growth [54]. The present results were consistent in that light and severe drought caused a reduction in soil nutrient contents. However, the addition of exogenous beneficial microorganisms can influence plant drought tolerance by affecting soil properties [17]. In the present study, soil nutrient contents, and activities of sucrase, phosphatase, and urease were significantly increased in response to *T. longibrachiatum* addition under drought stress. *Trichoderma* spp. increase soil nutrient cycling by secreting organic acids and various extracellular enzymes to dissolve minerals in the soil [13]. *Trichoderma asperellum* produces phosphatases that dissolve inorganic or organic phosphates [55]. Maeda et al. [56] reported that *Trichoderma* spp. decompose nitrogen compounds into available nitrogen and release less NO_2_. These results suggested that *T. longibrachiatum* might enhance the drought tolerance of *P. massoniana* by improving the soil nutrient availability. However, further study is required to identify metabolites secreted by *T. longibrachiatum* under drought stress that affect soil nutrient cycling.

Dry soil is not conducive to the survival of microbes in the plant rhizosphere. However, inoculation with exogenous beneficial microorganisms can affect the microbial community composition to improve plant stress tolerance [18]. In the present study, *T. longibrachiatum* application resulted in significant differences in the composition and diversity of the bacterial and fungal communities in the *P. massoniana* rhizosphere compared with those of the −T treatment and significantly enhanced the relative abundance of the dominant phyla, including Proteobacteria and Mortierellomycota, under drought stress. Proteobacteria are extremely important for global carbon, nitrogen, and sulfur cycling [57], which are vulnerable to drought stress [58]. Bu et al. [59] reported that drought stress reduced the relative abundance of Proteobacteria in a subtropical evergreen forest, whereas in the rainy season, Proteobacteria are enriched [60]. Proteobacteria have been identified in a “core microbiome” associated with drought-stressed roots [61]. These results suggested that Proteobacteria, enhanced under drought stress by *Trichoderma* inoculation, might play a crucial role in influencing the drought tolerance of *P. massoniana*. In addition, the dominant genera *Burkholderia*, *Rhodanobacter*, and *Trichoderma* were enriched in the +T treatment under drought stress. *Burkholderia* spp. have growth-promoting attributes and enhance plant tolerance of various abiotic stresses [62]. Naveed et al. [63] observed that *B. phytofirmans* PsJN increases maize biomass, leaf area, and RWC under drought stress while regulating cellular homeostasis and ROS detoxification at the transcription level [64]. Most *Burkholderia* spp. have nitrogen fixation, phosphate solubilization, potassium releasing, or phytohormone secretion abilities, which contribute to soil nutrient availability and plant growth [65,66]. In the current study, *Burkholderia* was positively correlated with soil AN content. *Rhodanobacter* spp. are strong competitors in acidic and anaerobic soils and may contribute to the majority of denitrification activity in acidic soils [67]. In the present study, *Rhodanobacter* was strongly correlated with AN and TN contents in the soil. Inoculation with certain *Trichoderma* spp. increases soil urease activity, and promotes soil AP and NH4^+^-N contents [17]. Similarly, it was observed that *Trichoderma* was positively correlated with soil urease activity, and AN and organic carbon contents. These results suggested that *T. longibrachiatum* treatment might effectively increase *Burkholderia*, *Rhodanobacter*, and *Trichoderma* abundance in the soil, synergistically regulating soil nutrient contents, and thereby enhancing the drought tolerance of *P. massoniana*.

The interactions between microorganisms have important influences on the stability and function of soil ecosystems [68]. To explore the critical dominant taxa promoted by *T. longibrachiatum* treatment, a microbial network analysis was conducted in the present research. *Trichoderma longibrachiatum* inoculation evidently changed the interaction network among microorganisms, especially for bacteria. Certain dominant taxa belonging to the genera *Penicillium*, *Trichoderma*, *Simplicillium*, *Saitozyma*, *Burkholderia*, *Bradyrhizobium*, *Sinomonas*, and *Mycobacterium* were strongly associated with other microorganisms in the +T treatment. Many *Penicillium* species supply various nutrients and phytohormones, and interact positively with roots of plants. *Penicillium resedanum* LK6 produces GA and improves pepper (*Capsicum annuum*) growth under drought stress [69]. Some *Simplicillium* spp. are utilized as biological control agents in plants [70]. *Saitozyma* is able to incorporate carbon from cellulose in dead plant biomass [71]. Egamberdieva et al. [72] reported that inoculation with biochar-based *Bradyrhizobium* enhanced growth as well as nitrogen and phosphorus uptake of lupin (*Lupinus angustifolius*) under drought stress. *Sinomonas* isolates screened from the rhizosphere of organically cultivated rice have high IAA production ability [73]. Karmakar et al. [74] observed that a *Mycobacterium* sp. had plant-growth-promoting traits and mitigated the effects of drought stress on rice. Based on the present results, the dominant taxa enriched in response to *T. longibrachiatum* inoculation may have crucial functions in enhancing soil nutrient availability, promoting plant growth, and improving the tolerance of *P. massoniana* to drought stress. However, further investigation is necessary to determine the interaction mechanism between *P. massoniana* and the rhizosphere microbial community.

In this study, the rhizosphere environment was closely related to the drought resistance of *P. massoniana* seedlings. Drought caused a decrease in soil fertility and microbial community diversity. However, the inoculation of beneficial *Trichoderma* not only significantly increased the contents of soil nutrients but also significantly increased the richness and diversity of fungal communities. *Trichoderma* treatment changed the interaction network among microorganisms in the rhizosphere soil. Some dominant bacterial taxa (*Burkholderia*, *Bradyrhizobium*, *Sinomonas*, and *Mycobacterium*, etc.) and fungal taxa (*Penicillium*, *Trichoderma*, *Simplicillium*, *Saitozyma*, etc.) may be important contributors to variations in soil nutrient. 

Based on the above results and analysis, a summary diagram was found that *T. longibrachiatum* inoculation changed plant physiology and soil environment under drought stress (Figure 8). Osmotic substance contents, antioxidant enzyme activities, and photosynthetic parameters were improved after *T. longibrachiatum* treatments under drought stress, which could regulate the osmotic balance [44], scavenge ROS [47], and enhance photosynthesis, respectively, ultimately reducing plant drought damage. Additionally, *T. longibrachiatum* inoculation influenced the rhizosphere soil microbial community, which may contribute to increased soil nutrients, synergistically promoting seedling growth under drought stress.

## 5. Conclusions

*Trichoderma longibrachiatum* effectively colonizes the roots of *P. massoniana* seedlings under drought stress and significantly enhances plant growth and drought tolerance by altering the plant physiology and the rhizosphere environment. Under drought stress, *T. longibrachiatum* treatment improves seedling height, root volume, shoot fresh weight, and phosphorus, potassium, and nitrogen content. *Trichoderma longibrachiatum* inoculation decreases MDA content and REC and increases antioxidant enzyme activity, osmotic substance content, and photosynthesis activity compared with those of non-inoculated seedlings under drought stress. In addition, *T. longibrachiatum* treatment markedly improves soil nutrient contents and enzyme activities and changes the composition of the soil bacterial and fungal communities. Based on microbial network analysis, certain dominant taxa driven by *T. longibrachiatum* inoculation, including genera of *Penicillium*, *Trichoderma*, *Simplicillium*, *Saitozyma*, *Burkholderia*, *Bradyrhizobium*, *Sinomonas*, and *Mycobacterium*, are indicated to play crucial roles in enhancing the drought resilience of *P. massoniana*.

## Figures and Tables

**Figure 1 jof-09-00694-f001:**
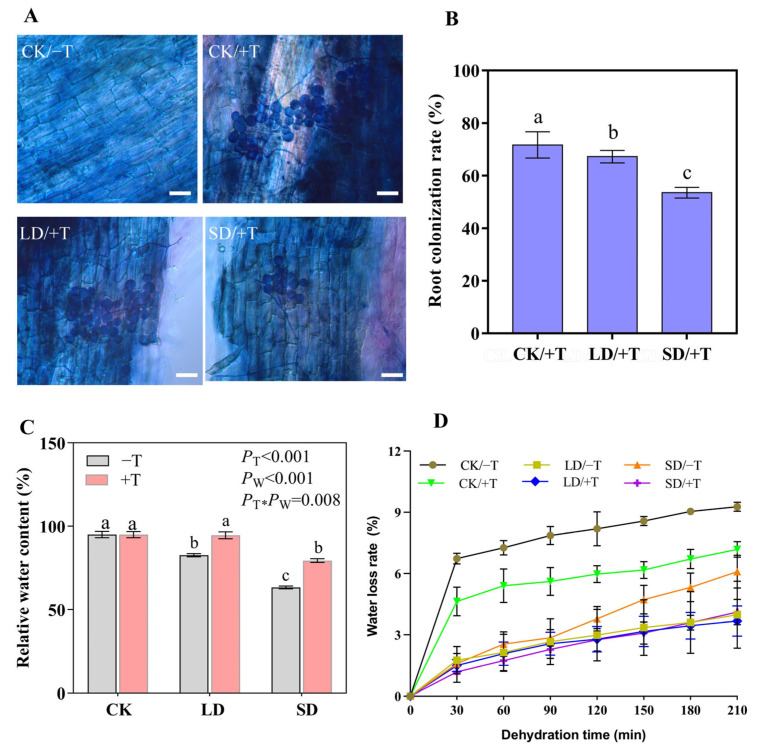
Determination of root colonization and water status of seedlings. (**A**) Root colonization observation. CK/−T, the roots without *T. longibrachiatum* (Tl) inoculation under well-watered conditions; CK/+T, the roots with Tl inoculation under well-watered conditions; LD/+T, the roots with Tl inoculation under light drought conditions; SD/+T, the roots with Tl inoculation under severe drought conditions. Bars, 25 μm. (**B**) The root colonization rate, (**C**) The relative water content of needles, (**D**) Water loss rate of needles. The *p*-value represents the significance level. T and W indicate *T. longibrachiatum* and water gradient treatments, respectively. Values of ±standard error were represented by error bars, and significant differences at *p* < 0.05 were indicated by different letters.

**Figure 2 jof-09-00694-f002:**
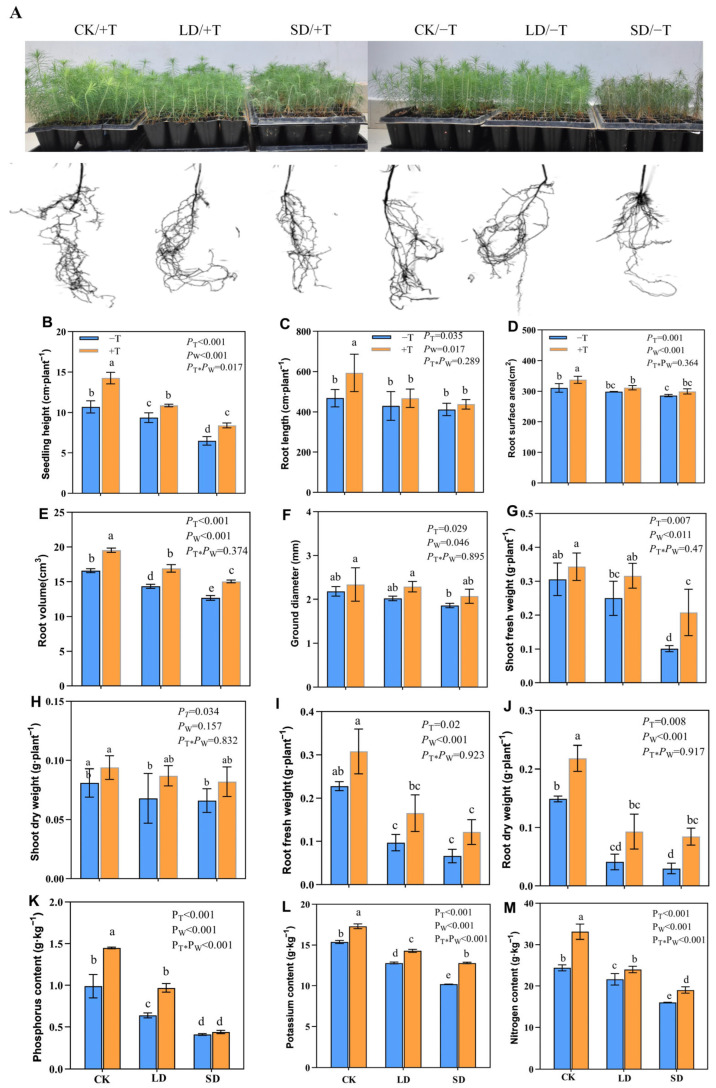
Effect of *T. longibrachiatum* and drought treatments on the growth and nutrient contents of masson pine seedlings. (**A**) Morphological observation of seedlings. (**B**–**J**) Determination of different growth parameters, including seedling height (**B**), root length (**C**), root surface area (**D**), root volume (**E**), ground diameter (**F**), shoot fresh weight (**G**), shoot dry weight (**H**), root fresh weight (**I**), and root dry weight (**J**). (**K**–**M**) Measurement of different nutrient elements, including phosphorus content (**K**), potassium content (**L**), and nitrogen content (**M**). CK, well-watered; LD, light drought; SD, severe drought; −T, the seedlings without *T. longibrachiatum* (Tl) inoculation; +T, the seedlings with Tl inoculation. *p*-values show what was not significant and significant for *p* > 0.05 and *p* < 0.05, T and W indicate *T. longibrachiatum* and water gradient treatments, respectively. Different letters indicate significant differences at the 0.05 level.

**Figure 3 jof-09-00694-f003:**
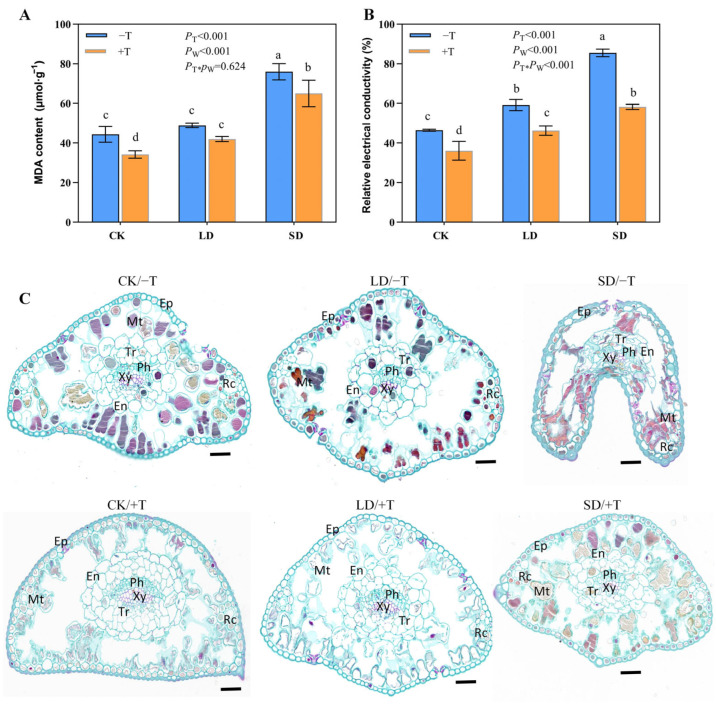
Effect of *T. longibrachiatum* and drought treatments on masson pine MDA content (**A**), relative electrical conductivity (**B**), and needles anatomical structure (**C**). Different letters indicate significant differences at the 0.05 level. Ep, epidermis; En, endothelium; Mt, mesophyll tissue; Rc, resin canal cavity; Xy, xylem; Ph, phloem; Tr, transport tissue; Bars, 50 μm. CK, well-watered; LD, light drought; SD, severe drought; −T, the seedlings without *T. longibrachiatum* (Tl) inoculation; +T, the seedlings with Tl inoculation.

**Figure 4 jof-09-00694-f004:**
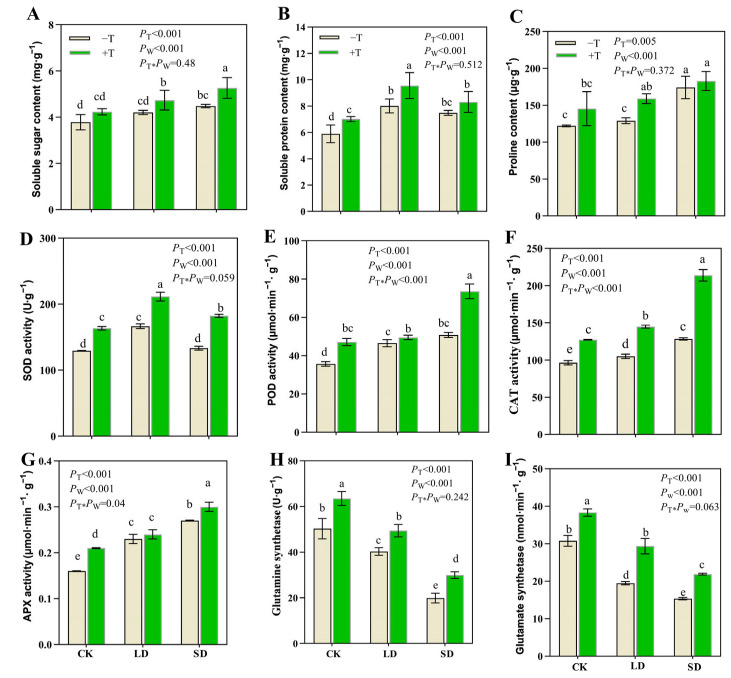
Effect of *T. longibrachiatum* and drought treatments on osmotic substances and antioxidant enzyme activities in *P. massoniana* seedlings. (**A**) Soluble sugar content, (**B**) Soluble protein content, (**C**) Proline content, (**D**) SOD activity, (**E**) POD activity, (**F**) CAT activity, (**G**) APX activity, (**H**) Glutamine synthetase, (**I**) Glutamate synthetase. *p*-values show what was not significant and significant for *p* > 0.05 and *p* < 0.05, T and W indicate *T. longibrachiatum* and water gradient treatments, respectively. Different letters indicate significant differences among all treatments at the 0.05 level.

**Figure 5 jof-09-00694-f005:**
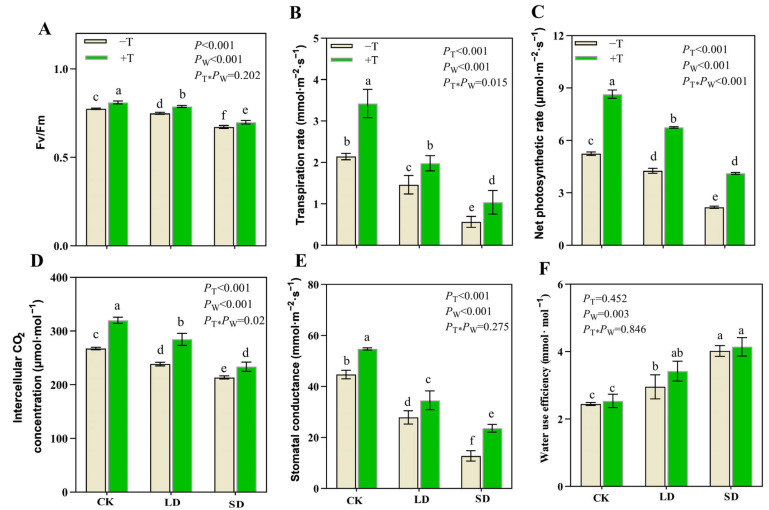
Analysis of photosynthetic parameters under *T. longibrachiatum* and drought treatments. (**A**) Fv/Fm, (**B**) Transpiration rate (Tr), (**C**) Net photosynthetic rate (Pn), (**D**) Intercellular CO_2_ concentration (Ci), (**E**) Stomatal conductance (Gs), (**F**) Water use efficiency (WUE). The *p*-value represents the significant level. T and W indicate *T. longibrachiatum* and water gradient treatments, respectively. Bars not sharing the same lowercase letters are significantly different among treatments at the 0.05 level.

**Figure 6 jof-09-00694-f006:**
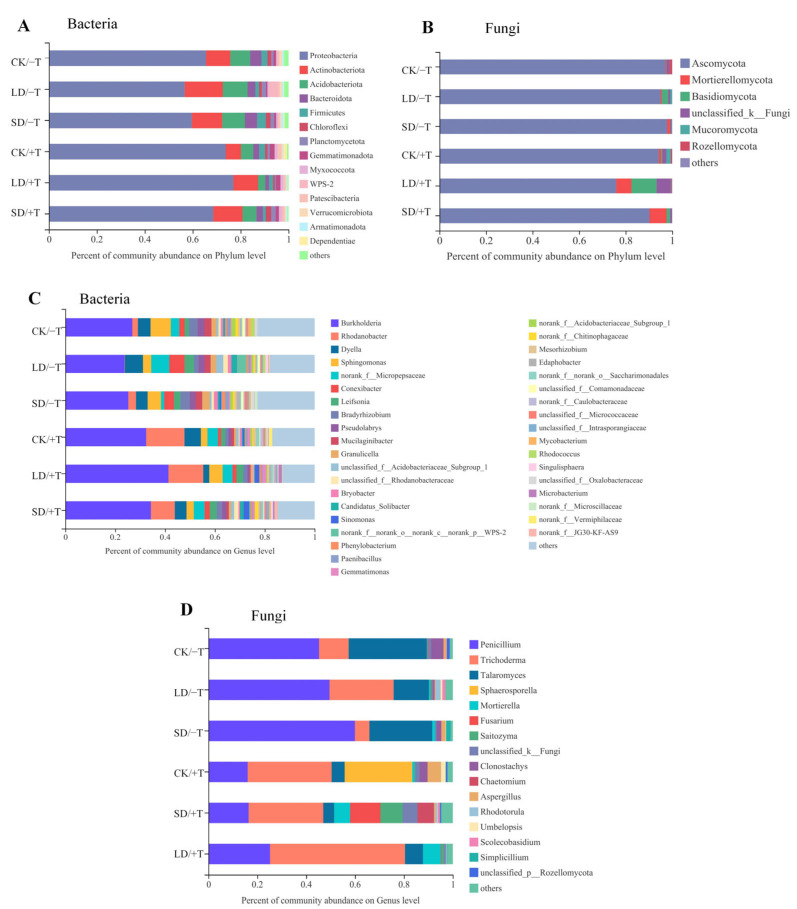
Relative abundances of bacterial (**A**) and fungal (**B**) phyla, and the bacterial (**C**) and fungal (**D**) genus in the rhizosphere soil of *P. massoniana* treated with *T. longibrachiatum* and drought treatment. CK/−T, seedlings without *T. longibrachiatum* (Tl) inoculation under well-watered conditions; LD/−T, seedlings without Tl treatment under light drought conditions; SD/−T, seedlings without Tl under severe drought conditions; CK/+T, seedlings with Tl inoculation under well-watered condition; LD/+T, seedlings with Tl treatment under light drought condition; SD/+T, seedlings with Tl under severe drought conditions. The abscissa is the proportion of the community abundance of species in the sample. Columns of different colors represent different species, and the length of the columns represents the proportion of species.

**Figure 7 jof-09-00694-f007:**
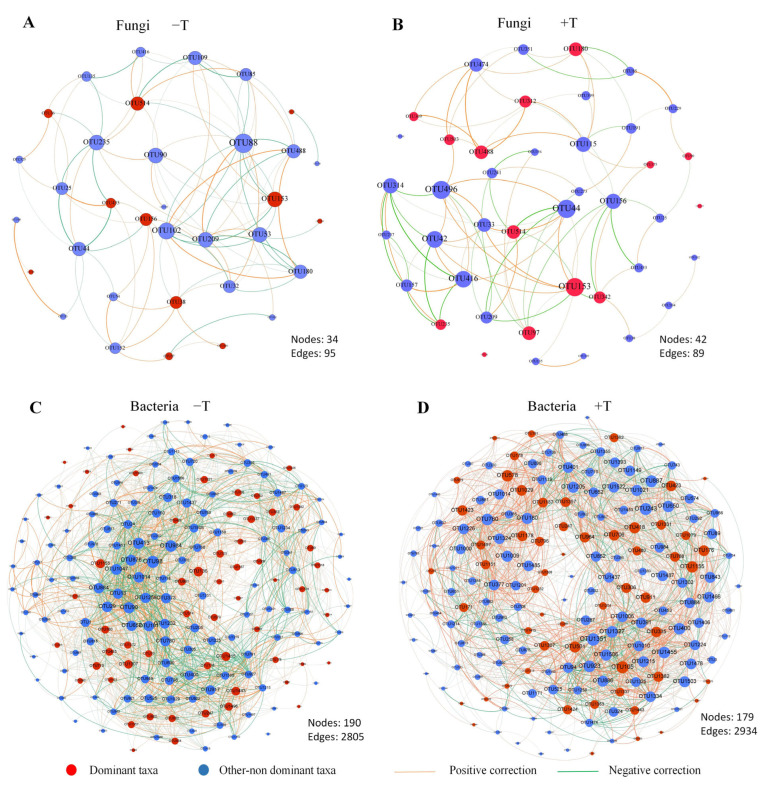
Network analysis showing the correlation between microorganisms based on Spearman’s correlation analysis. Fungal (**A**,**B**) and bacterial (**C**,**D**) co–occurrence networks between dominant and other-non dominant taxa under “−T” and “+T” treatments. “−T” indicates all samples without *T. longibrachiatum* (Tl) inoculation, and “+T” indicates all samples with *T. longibrachiatum* (Tl) inoculation. Nodes represent OTUs; red notes represent dominant taxa, blue represents other non-dominant taxa. The size of each node was proportional to the degree of connections; edges represent Spearman’s correlation; The orange and green lines represented the strongly positive (*R* > 0.6, *p* < 0.05) and negative (*R* < −0.6, *p* < 0.05) relationships, respectively.

**Figure 8 jof-09-00694-f008:**
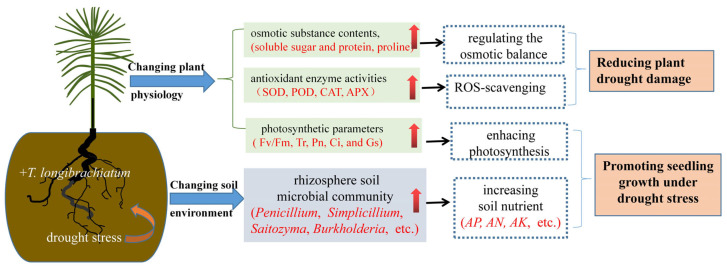
Summary diagram of the effects of *T. longibrachiatum* inoculation on stimulating growth and enhancing drought resistance of *P. massoniana* seedlings. The red arrow represents rising.

## Data Availability

The data in this study are available on request from the corresponding author/first author.

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
