# Peer review of "Trichoderma longibrachiatum Inoculation Improves Drought Resistance and Growth of Pinus massoniana Seedlings through Regulating Physiological Responses and Soil Microbial Community"

_jof, 2023, doi:10.3390/jof9070694_

Round 1
Reviewer 1 Report
The article presents influence of T. longibrachiatum on early growth stages of P. massoniana under drought conditions. The beneficial interactions are demonstrated through a series of experiments including plant and soil attributes, and microbiome structure. Following points may be considered in the revised manusrcitp -
- The selection of soil medium 6:3:1 of pearlite:vermiculite:soil may be elaborated
- Three seedlings were accommodated in each pot, The pot dimensions are not mentioned
- Inoculum size 10^6 spores in 30mL X 2 times (20mL/seedling) appears too high. This could be difficult in a large scale scenario of real-field implementation.
- Plant growth conditions during pre and post inoculation phase are not mentioned
- A correlation analysis of the measured physicochemical attributes may be included to understand the interrelated expression of the attributes.
- Soil attributes are measured post-treatment. However, the status of soil characteristics at T0 would further help to understand the treatment-influence. This is important as the initial nutrient availability in the growth substrate may also influence indigenous stress tolerance in plants.
- Discussion could also include a paragraph to integrate the major findings on plant, soil, microbiome, and the treatments.
Author Response
Dear Reviewer,
Thank you for your good work on our manuscript, and we are very grateful to Reviewer’s suggestion. Based on these comments and suggestions, we have carefully revised our original manuscript. We also responded point by point to each reviewer’s comments, along with a clear indication of the location of the revision. Hope these will make it more acceptable for publication.
Response to review 1#
1. The selection of soil medium 6:3:1 of pearlite: vermiculite: soil may be elaborated
Response: According to the suggestion, the selection of soil medium has been elaborated, and we added the source and background information of the soil (line 95-101).
2. Three seedlings were accommodated in each pot, The pot dimensions are not mentioned
Response: The pot dimensions have been supplemented (line 101-102).
3. Inoculum size 10^6 spores in 30mL X 2 times (20mL/seedling) appears too high. This could be difficult in a large scale scenario of real-field implementation.
Response: We apologize that we did not consider the actual field experiment application. In the future experiment, we will consider increasing the spore concentration to 1 ×10^8 spores, reducing the spore inoculation volume, conveniently in a large scale scenario of real-field implementation.
4. Plant growth conditions during pre and post inoculation phase are not mentioned
Response: According to the suggestion, plant growth conditions during pre and post inoculation phase have been added (line 103-105, line 116-117)
5. A correlation analysis of the measured physicochemical attributes may be included to understand the interrelated expression of the attributes.
Response: We had added a correlation analysis of the measured soil physicochemical attributes as Table S3, and the analysis description was presented in the Result 3.6. (line 360-365).
6. Soil attributes are measured post-treatment. However, the status of soil characteristics at T0 would further help to understand the treatment-influence. This is important as the initial nutrient availability in the growth substrate may also influence indigenous stress tolerance in plants.
Response: According to the suggestion, we had added the soil background information before inoculation treatment (line 98-101).
7. Discussion could also include a paragraph to integrate the major findings on plant, soil, microbiome, and the treatments.
Response: we had added a summary diagram as Figure 8 and included a paragraph to integrate the major findings on plant, soil, microbiome, and the treatments (line 581-592).

Reviewer 2 Report
This paper focused on the effect of Trichoderma longibrachiatum on rhizosphere microbial diversity and physiological change of Pinus massoniana under drought stress. The authors inoculated Trichoderma longibrachiatum to Pinus massoniana for 60 days and then applied three different drought treatments for 15 days. They observed the effects of Trichoderma longibrachiatum on the growth, hormone contents, water content, tissue structure, nutrient content, activities of antioxidant enzyme and soil enzyme, photosynthesis, and rhizosphere microbial diversity of Pinus massoniana under drought stress.
The experiments were conducted well and contain a lot of information. However, I was confused about why the authors investigated the microbial composition of the sterilized medium. Furthermore, this study showed contribution of Trichoderma longibrachiatum from photosynthesis, nutrient, hormone, tissue structure, rhizosphere microbial diversity and enzyme aspects, but it's hard to organize them together. The authors have described a lot of information of plants under drought stress and provided a lot of possibilities but didn't clarify them clearly. It would be better if the authors focus on a few aspects and explain them comprehensibly. This manuscript would be publishable once the comments are addressed properly.
Here are some comments:
Introduction
1Line 67-70: This sentence may need to be revised.
2Line 71-72: "In our previous research, four endophytic Trichoderma strains were isolated from P. massoniana and their drought tolerance was evaluated …". How did you evaluate their drought tolerance? Is there any reference that can be cited?
Methods
1. Line 99: “The spore suspension was adjusted to 106 CFU/mL…". How did the authors measure the spore suspension as 106 CFU/mL? It will be great if the authors can add more information in detail.
2Line 92: How do the authors sterilize the medium for plant growth? Why did the authors investigate the microbial composition of the sterilized medium?
Results
1Figure 1A: The spores in root are not clear enough. The authors didn't describe the difference of these four pictures in Figure 1A. Is that necessary to put this figure here?
2Figure 3: These pictures were conducted well. that possible to show the difference of drought damage to leaf protection cell among these treatments? Since the leaf protection cell may change in plants under drought stress, it would be interesting to show that.
3There were some P values written in uppercase and some written in lowercase. Some of them were italic but some of them were not. Please revise them as one format.
Discussion
1. What is the relationship between antioxygen enzymes and the photosystem damage caused by ROS?
2Line 468: How does drought stress reduce chlorophyll content? Is there any connection with the tissue structure damage by drought stress?
3How does hormone metabolism be changed by drought stress in this study? Is there any relationship among hormone metabolism, ROS, and microbial composition?
In general, it’s a well written paper but it didn’t explain the result clearly so that major revision is needed.
Author Response
Dear Reviewer,
Thank you for your good work on our manuscript, and we are very grateful to Reviewer’s suggestion. Based on these comments and suggestions, we have carefully revised our original manuscript. We also responded point by point to each reviewer’s comments, along with a clear indication of the location of the revision. Hope these will make it more acceptable for publication.
Response to 2#
Introduction
1. Line 67-70: This sentence may need to be revised.
Response: According to the suggestion, this sentence have been revised (line 67-69).
2. Line 71-72: "In our previous research, four endophytic Trichoderma strains were isolated from massoniana and their drought tolerance was evaluated …". How did you evaluate their drought tolerance? Is there any reference that can be cited?
Response: Yes, we have added the reference in line 75.
“The reference described that four Trichoderma species,including T. koningiopsis,T. virens, T. spirale and T. longibrachiatum, were grown on PDA plates contained different concentrations of PEG-6000, then sporulation capacity, mycelial biomass, antioxidant enzyme activities, and osmolytes contents of different Trichoderma species were used to evaluate drought resistance capacity.”
Ref.
Rui, Z., Jiang, X., Yu, C. Effects of polyethylene glycol 6 000 stress on the growth and physiology of Trichoderma longibrachiatum. Journal of Forest and Environment 2023, 43, 210-216. https://doi.org/10.13324 /j. cnki. jfcf. 2023.02.013. (In Chinese)
Methods
1. Line 99: “The spore suspension was adjusted to 106CFU/mL…". How did the authors measure the spore suspension as 106 CFU/mL? It will be great if the authors can add more information in detail.
Response: According to the suggestion, we have added the method for measuring the spore suspension as 106 CFU/mL (line 110-112).
2. Line 92: How do the authors sterilize the medium for plant growth? Why did the authors investigate the microbial composition of the sterilized medium?
Response: In order to rule out the possible interference by original microorganisms in the rhizosphere soil of P. massoniana seedlings, the medium for plant growth was sterilized, and can directly prove that the inoculation of exogenous T. longibrachiatum induced the formation of a unique rhizosphere microbial community, then affected the growth and drought resistance of seedlings. This method refers to the method described by Halifu et al. (2019) and Li et al. (2022).
Ref.
Halifu, S., Deng, X., Song, X., Song, R. Effects of two trichoderma strains on plant growth, rhizosphere soil nutrients, and fungal community of Pinus sylvestris var. mongolica annual seedlings. Forests 2019, 10, 758. https://doi.org/10.3390/f10090758
Li, M., Ren, Y., He, C., Yao, J., Wei, M., He, X. Complementary effects of dark septate endophytes and Trichoderma strains on growth and active ingredient accumulation of astragalus mongholicus under drought stress. J. Fungi 2022, 8, 920. https://doi.org/10.3390/jof8090920.
Results
1. Figure 1A: The spores in root are not clear enough. The authors didn't describe the difference of these four pictures in Figure 1A. Is that necessary to put this figure here?
Response: We apologize that the spores in root are not clear enough, because it was needed to verify whether T. longibrachiatum could successfully colonize the root of seedlings under normal and drought conditions after inoculation, the morphology of spores and hyphae were observed by a light microscope. In future experiments, we will delve into the colonization mechanism of T. longibrachiatum, and more clear spores and hyphae will be observed using a scanning electron microscopy. At the same time, we have added the description of the differences among the four graphs in Figure 1A in the Results 3.1. (line 244-248).
2. Figure 3: These pictures were conducted well. that possible to show the difference of drought damage to leaf protection cell among these treatments? Since the leaf protection cell may change in plants under drought stress, it would be interesting to show that.
Response: Yes, these pictures could show the difference of drought damage to leaf protection cell among these treatments. Under well-watered (CK) treatment, the cell size was uniform; mesophyll and endodermal cells were abundant, evenly, tightly, and orderly arranged. But under drought stress, the mesophyll cells was severely deformed and shrunken. In future experiments, in order to deeply investigate the damage of cells in response to drought stress, a scanning electron microscopy will be used to observe the ultra-structural changes of each cell, including organelle and cytoplasmic inclusions.
3. There were some P values written in uppercase and some written in lowercase. Some of them were italic but some of them were not. Please revise them as one format.
Response: According to the suggestion, we have revised them as one format (line 231, 236, 266, 286, 334, 435).
Discussion
1. What is the relationship between antioxygenenzymes and the photosystem damage caused by ROS?
Response: The activity of antioxidant enzymes will reduce the accumulation of ROS, thereby reducing the damage of ROS to the photosynthetic system, we have added the discussion as follows (line 492-497):
“Excessive ROS can impair the chloroplasts, decrease the photochemical reactions, and finally suppress the photosynthesis and yield of the crop (Huang et al. 2019). Plants have developed several enzymatic antioxidant protection mechanisms to counteract the damaging effects of ROS under environmental stresses. Photosystem also could evolve a highly efficient antioxidant defense system, including an enzymatic scavengers (SOD and CAT), reducing ROS accumulation (Pospíšil, 2012).”
Ref:
Huang, B., Chen, Y.E., Zhao, Y.Q., Ding, C.B., Liao, J.Q., Hu, C., Zhou, L.J., Zhang, Z.W., Yuan, S., and Yuan, M. Exogenous melatonin alleviates oxidative damages and protects photosystem II in maize seedlings under drought stress. Front. Plant Sci. 2019 10, 677. https://doi.org/10.3389/fpls.2019.00677
Pospíšil, P. Molecular mechanisms of production and scavenging of reactive oxygen species by photosystem II. BBA-Bioenergetics 2012, 1817, 218-231. https://doi.org/10.1016/j.bbabio.2011.05.017.
2. Line 468: How does drought stress reduce chlorophyll content? Is there any connection with the tissue structure damage by drought stress?
Response: There are some connections between chlorophyll content reduction and the tissue structure damage by drought stress, we have added the discussion as follows:
“Drought stress could disrupt tissue structure, leading to cell deformation, rupture and even death. After cell death, chlorophyll would be released from the chloroplast, and free chlorophyll was unstable and easily degraded (Sun et al., 2020).” (line 499-501)
Ref:
Sun, Y., Lu, R., Pan, L., Wang, X., Tu, K. Assessment of the optical properties of peaches with fungal infection using spatially-resolved diffuse reflectance technique and their relationships with tissue structural and biochemical properties. Food Chem 2020, 321, 126704. https://doi.org/10.1016/j.foodchem.2020.126704.
3. How does hormone metabolism be changed by drought stress in this study? Is there any relationship among hormone metabolism, ROS, and microbial composition?
Response: Previous studies have shown that drought will change the plant hormone content, but how drought affects the hormone metabolism pathway may require transcriptome analysis. Since there is no relevant validation test in this paper, and in order to focus on some key aspects and explain them comprehensibly, we deleted the results about hormone metabolism, meanwhile we added a summary diagram as Figure 8 and included a paragraph to organize the major findings on plant, soil, microbiome, and the treatments.

Round 2
Reviewer 2 Report
Thank you for addressing the suggested edits. In Lines 91-99, the authors added the physicochemical properties of mixed media. They have mentioned that soil was collected from P. massoniana plantation. However, the sterilization methods are still not mentioned. Is that mean the soil wasn't sterilized? They may need to add related information in the introduction/method to describe why they use the sterilized/ unsterilized soil.
It’s a well written paper.
Author Response
Response: Thank you for your good work on our manuscript, according to the suggestion, we had added the sterilization methods in Line 100, and the soil was sterilized and the related information to describe why we use the sterilized soil was added in the Method 2.1 (Line 97-101). Meanwhile, we have made appropriate modifications to the language.